# Role of Glycans in Equine Endometrial Cell Uptake of Extracellular Vesicles Derived from Amniotic Mesenchymal Stromal Cells

**DOI:** 10.3390/ijms26041784

**Published:** 2025-02-19

**Authors:** Giulia Gaspari, Anna Lange-Consiglio, Fausto Cremonesi, Salvatore Desantis

**Affiliations:** 1Laboratory of Reproduction and Regenerative Medicine, Department of Veterinary Medicine and Animal Science (DIVAS), University of Milan, 26900 Lodi, Italy; anna.langeconsiglio@unimi.it (A.L.-C.); fausto.cremonesi@unimi.it (F.C.); 2Department of Precision and Regenerative Medicine and Ionian Area, University of Bari Aldo Moro, 70010 Valenzano, Italy; salvatore.desantis@uniba.it

**Keywords:** extracellular vesicles, lectins, carbohydrates, glycans, equine

## Abstract

Extracellular vesicles (EVs) are important mediators of cell–cell communication thanks to their ability to transfer their bioactive cargo, thus regulating a variety of physiological contexts. EVs derived from amniotic mesenchymal/stromal cells (eAMC-EVs) are internalized by equine endometrial cells (eECs) with positive effects on regenerative medicine treatments. As the cellular uptake of EVs is influenced by the glycan profile of both EVs and target cells, this study is focused on the role of surface glycans in the uptake of eAMC-EVs by recipient eECs. Equine ECs were obtained by enzymatic digestion of uteri from healthy mares. Equine AMC-EVs were isolated from amniotic cell cultures according to a standardized protocol. The glycan pattern was studied using a panel of lectins in combination with fucosidase and neuraminidase treatment. Both eECs and eAMC-EVs expressed N-linked high mannose glycans, as well as fucosylated and sialylated glycans. All these glycans were involved in the uptake of eAMC-EVs by eECs. The internalization of eAMC-EVs was strongly reduced after cleavage of α1,2-linked fucose and α2,3/α2,6-linked sialic acids. These results demonstrate that surface glycans are involved in the internalization of eAMC-EVs by eECs and that fucosylated and sialylated glycans are highly relevant in the transfer of bioactive molecules with effects on regenerative medicine treatments.

## 1. Introduction

In equine species, EVs have been isolated from mesenchymal-derived stem cell cultures of tissues such as bone marrow [1,2], adipose tissue, synovial fluid [1], and amniotic mesenchymal stromal cells (eAMCs) [3,4]. A positive effect of eEVs in the reduction of inflammation in damaged tendon cells [3], endometrial cells [4,5], and chondrocytes [1] has been reported in vitro.

Glycans are complex carbohydrates abundant on the cell surface and are involved in cellular interactions by recognizing specific glycan-binding proteins [6]. The surface of EVs is highly glycosylated [7,8,9], and this glycosylation is important for their uptake by recipient cells [10,11,12,13]. Indeed, the glycan profile of both EVs and the acceptor cells strongly influences the cellular uptake behavior of EVs [14]. Thus, the interaction of EVs with specific receptors located on the surface of the target cells plays a significant role in the mode of transference of molecules carried by EVs.

Most surface glycans belong to N- and O-glycan families. N-glycans are characterized by the attachment of the glycan chain to an asparagine residue of a protein, whereas in O-glycans, the glycan chain binds to a threonine or serine residue of a protein. The surface glycans of EVs and target cells could terminate with sialic acid and fucosylated residues. Sialoglycans are considered important in the uptake of EVs [11,13,14,15,16]. Fucosylated glycans also contribute to EV uptake by recipient cells [12,13].

Lectins are proteins that selectively bind to specific carbohydrate structures and recognize both N-glycans and/or O-glycans, including galactose, N-acetylgalactosamine, N-acetyglucosamine, fucose, mannose, and sialic acid. Furthermore, they are useful tools for distinguishing sugar isomers, linkage, and branching of glycans [17,18]. Lectins are widely used for the in situ investigation of cell and EV surface glycopatterns [9,11,12,14,19,20,21,22].

Extracellular vesicles from different sources may have some glycans in common [20] or display glycans specific to the cell type [14,19].

The glycopattern of eAMCs and their derived EVs (eAMC-EVs) has been previously analyzed by Desantis et al. [21], who demonstrated that eAMC-EVs were enriched in some N- and O-linked glycans when compared to their parent cells.

It has been proven that eAMC-EVs can be incorporated by equine tendon cells [3] and endometrial cells (eECs) [4] in vitro. In vivo, eAMC-EVs have been used for regenerative medicine treatment of a mare affected by chronic degenerative endometritis [5] and to prevent persistent post-breeding endometritis [23].

Considering the in vivo regenerative effects of eAMC-EV, it may be hypothesized that eAMC-EV molecules are transferred into eECs. The purpose of this study was to understand which surface molecules on both EV and eEC membranes are involved in this interaction.

Extracellular vesicle internalization occurs by their endocytic uptake by target cells or through direct fusion with cell membrane; EV surface glycoproteins and receptors are essential for this process [9]. Thus, the present study intended to assess the glycan pattern of eAMC-EVs and eECs and to verify their involvement in the uptake process. The effect of masking the eAMC-EV surface glycans and de-fucosylation and de-sialylation of eAMC-EVs and eECs on the incorporation of eAMC-EVs into eECs was also studied.

## 2. Results

### 2.1. Evaluation of eAMC-EV Incorporation into eECs

Initially, the effective incorporation of eAMC-EVs into eECs was assessed by labeling vesicles with PKH-26, a red fluorescent chromophore that was detected by fluorescence microscopy and confocal microscopy. In all the samples studied, the fluorescence signal due to the incorporation of eAMCV-EVs into eECs was detectable. Nuclei were identified with Hoechst 33342 (blue), and eAMCV-EVs labeled with PKH-26 (red) were visible inside the cell (Figure 1).

### 2.2. Characterization of AMC-EV Glycans with a Panel of FITC-Labeled Lectins

In order to analyze the glycan pattern of eAMC-EVs, a panel of nine FITC-labeled lectins was used (Table 1).

Equine AMC-EVs labeled with PKH-26 (red) showed positivity to the nine tested lectins labeled with FITC (green). When the two fluorescence channels were merged, the positivity appeared as yellow/orange (Figure 2).

The control is represented by eAMC-EVs only, not conjugated with lectins.

### 2.3. Detection of N-Linked, Fucosylated, and Sialylated Glycans on eEC Surface with FITC-Labelled Lectins

To detect the presence of N-linked, sialylated, and fucosylated glycans on the surface of eECs, they were incubated with the lectins Con A, AAL, MAL II, and SNA and observed under a light photomicroscope.

The fluorescence images showed that eECs have binding sites on their surface membrane for the lectins included in the study, indicating the presence of high-mannose N-glycans (Con A), α1,2/α1,3/α1,4/α1,6-linked fucose (AAL), a2,3- and α2,6-linked sialic acids (MAL II and SNA, respectively) (Figure 3A–D).

After eAMC-EV uptake by eECs (Figure 4A,D,G,J), endometrial cells still display reactivity to AAL, Con A, MAL II, and SNA (Figure 4B,E,H,K). The merged images are shown in (Figure 4C,F,I,L).

### 2.4. Effect of De-Fucosylation and De-Sialylation on eEC Uptake by eAMC-EVs

In experiment 3, cultured eECs displayed different reactivity to the different lectins, showing the highest staining signal with Con A and decreasing signals with AAL, SNA, and MAL II (Figure 5).

To explore the possible role of fucosylated and sialylated glycans on eAMC-EV uptake by eECs, the effect of their enzymatic removal with fucosidase or neuraminidase, respectively, was evaluated. De-fucosylation significantly reduced the reactivity of eECs for AAL and SNA. Conversely, it revealed additional MAL II binding sites.

The removal of the fucose residues by fucosidase from eECs strongly decreased the AAL reactivity (Figure 6A–C); however, the reactivity for Con A, MAL II, and SNA was unchanged (Figure 5 and Figure 6D,G,J). When de-fucosylated eECs were incubated with eAMC-EVs, intracellular eAMC-EVs were not detected in AAL and Con A staining experiments (Figure 6B,E). However, uptake of eAMC-EVs was observed in de-fucosylated eECs incubated with MAL II and SNA (Figure 6H,K). In addition, the incorporation of eAMC-EVs was observed only in Con A binding cells (Figure 6F) and not in AAL, MAL II, and SNA-reactive eECs (Figure 6C,I,L) when normal or de-fucosylated eECs were incubated with fucosidase-treated eAMC-EVs.

The cleavage of sialic acid from eECs did not affect AAL and Con A reactivity (Figure 5 and Figure 7A,D), while it strongly decreased eEC binding sites for MAL II and SNA (Figure 7G,J). Intracellular eAMC-EV uptake was not seen after incubation of the de-sialylated eECs with eAMC-EVs (Figure 7B,H,K) except in Con A stained eECs (Figure 7E). The incubation of eECs with de-sialylated eAMC-EVs, as well as de-sialylated eECs with de-sialylated eAMC-EVs, did not result in eAMC-EV incorporation (Figure 7C,F,I,L).

Treatment with fucosidase and neuraminidase caused the failure of nuclear staining of several eECs with DAPI (Figure 6 and Figure 7).

### 2.5. Evaluation of eAMC-EV Incorporation into eECs After Masking of Carbohydrate Residues with Unconjugated Lectins

The panel of nine lectins previously tested in experiment 2 was used again to evaluate whether the corresponding sugar residues were also involved in eAMC-EV incorporation into eECs.

PKH-26-labeled eAMC-EVs were pre-incubated with unconjugated lectins. Subsequently, eECs were incubated with these eAMC-EVs. The masking of sugar residues on the eAMC-EV membrane prevented eAMC-EV uptake by eECs. Only Hoechst-labeled cell nuclei are visible inside the eECs.

By binding to vesicle glycoproteins, unlabeled lectins prevent the entry of eAMC-EVs (Figure 8A,B).

## 3. Discussion

Communication between cells is fundamental for all multicellular organisms, and information is exchanged through the secretion of soluble or non-soluble factors.

Most eukaryotic cells release EVs by cell membrane budding or by exocytosis of multivesicular cytoplasmatic bodies.

Extracellular vesicles are composed of a non-replicating lipid bilayer and contain many molecules, including nucleic acids (DNA, RNA, and microRNAs), proteins, and lipids. This cargo allows these nanoparticles to act as intercellular messengers capable of influencing physiological processes [24].

The interaction between eECs and eAMC-EVs has been detected in vitro [4] and in vivo [5,23]. Equine eAMC-EVs, including microRNA cargo, have been previously characterized [23,25] according to the MISEV guidelines [24]. In addition, in vitro, uptake of eAMCs by eECs was detected and evaluated by our research group using confocal microscopy [4].

It is well known that the EV surface includes a biomolecular corona containing several types of glycans [10]. In this study, we investigated whether glycans are involved in eAMC-EV-mediated communication with eECs.

Initially, surface glycans of eAMC-EVs were studied using a panel of nine lectins. Lectins are naturally occurring bioactive proteins that selectively bind carbohydrate molecules. Binding with high specificity and affinity to monosaccharides and oligosaccharides of complex carbohydrates present in solutions, on cell surfaces, in subcellular organelles, and in tissue sections, they are involved in several biological, biochemical, and immunochemical activities [26,27,28]. Lectins are useful for exploring tissue and cell glycosylation [17,18,29].

Although only a few lectins were chosen for this study, it was possible to identify in situ the presence of the main classes of glycans on the surface of EVs, such as high-mannose N-linked glycans with Con A; neutral O-linked glycans with PNA (considered a T-antigen, namely terminal Gal*β*1,3GalNAc binder); *α*2,3-sialylated O-linked glycans with MAL II; *α*2,6-sialylated N-acetyllactosamine (LacNAc) with SNA; fucosylated glycans with AAL, LTA, and UEA I; and terminal LacNAc and GalNAc with RCA_120_ and SBA, respectively [29].

Positive results were detected for all the lectins tested, indicating the presence of all the corresponding carbohydrates on the surface membrane of eAMC-EVs in all microscopic images: the yellow/orange fluorescence due to the combination of FITC-conjugated lectins (green) with eAMC-EVs labeled with PKH-26 (red) is evident.

These findings agree with a previous study performed with a microarray method for in situ glycoprofiling of eAMC-EVs [21]. Compared to the previous investigation, AAL was also used in this study. In this way, a wider spectrum of fucosylated glycans was detected, as AAL, in addition to α1,2- and α1,3-linked fucose (UEA I and LTA affinity), also binds α1,4- and α1,6-linked fucose [29]. Considering that EVs display some cell-specific variability in glycosylation [11,14,20], our results are in line with most reports on this topic. High mannose, complex N-linked glycans, lactosamine, α2,3/6 sialic acids, and fucose are the most commonly expressed glycan epitopes in EVs [11,19,20,22,30,31]. Although the roles of glycans are not well known, N-glycosylation is considered important for protein stability and modulates the availability of surface receptors [32]. Lactosamine and the high mannose concentration in EVs have been associated with glycoprotein sorting [33]. Sialoglycans provide a negative charge to the glycocalyx [34,35]. Finally, fucose epitopes have been found in tetraspanin CD63, an EV marker [31].

The goal of this study was to explore the involvement of glycans in the uptake of eAMC-EVs by eECs, and the presence of the more typical glycans in these cells was also investigated. Thus, eECs were incubated with Con A, AAL, MAL II, and SNA to show the presence of high-mannose N-glycans, α1,2/α1,3/α1,4/α1,6-fucosylated glycans and α2,3-/2,6-sialoglycans, respectively.

High-mannose N-glycans, fucosylated, and α2,3-/2,6-sialoglycans are commonly found on the surface of cultured cells [21,22,36,37], including eAMCs [21]. N-glycosylation and sialic acids are involved in many biological functions, including cell recognition, receptor activation, cell signaling, cell adhesion, and cell proliferation [38,39,40,41]. Moreover, fucosylation contributes to cell adhesion and signaling events [42,43,44].

Fluorescent micrographs demonstrate that eECs display binding sites for these four lectins, indicating that high-mannose N-glycans, α1,2/α1,3/α1,4/α1,6-fucosylated glycans, and α2,3-/2,6-sialoglycans are present on the cell membrane of eECs.

Incubation of eAMC-EVs with eECs resulted in the internalization of the vesicles into the eECs, and the positive reactivity to four lectins was maintained after this incorporation. However, the pre-incubation of eAMC-EVs with any of the nine lectins eliminated the uptake of eAMC-EVs by eECs. This indicates that the glycans detected in eAMC-EVs in this study take part in the process of incorporation of eAMC-EVs into eECs. Only the nucleus of the cells, stained by Hoechst, was visible by fluorescent microscope after internalization.

To better understand the role of fucose- and sialo–glycans in the interaction and incorporation of eAMC-EVs, we studied the effect of their removal on the incorporation of eAMC-EV by eECs. Fucosidase and neuraminidase enzymes were used to digest fucose and sialic acid residues, respectively.

De-fucosylation significantly reduced the reactivity of eECs for AAL. It has been reported that cell fucosylation can impact EV uptake [12]. In the present investigation, the incubation of de-fucosylated eECs with eAMC-EVs showed no sign of intracellular AMC-EVs not only in eECs incubated with AAL but also in those stained with Con A. The latter result might be the effect of three-dimensional modification of the N-linked glycan constituting the eEC receptor for eAMC-EV as a consequence of de-fucosylation. When de-fucosylated eAMC-EVs were incubated with normal or fucosidase-treated eECs, their uptake was only detected in a few Con A-binding cells. This suggests that the removal of fucose from eAMC-EVs may expose hidden cell receptors containing high-mannose glycans. Li et al. [12] reported that the pattern of cellular uptake of highly fucosylated-EVs was significantly different from EVs with low fucosylation. They also demonstrated fucosylated-EVs using the lectin LCA that binds α1,6-linked fucose [29] in hepatocellular carcinoma samples. The presence of α1,6-linked fucose (LCA and PSA reactivity) has also been found on the surface of EVs from AML12 and MLP29 hepatic murine cell lines [11]. Interestingly, in the present study, we indirectly detected α1,2-linked fucose in eECs and eAMC-EVs because the fucosidase used hydrolyzed α1,2-linked fucose residues. These findings suggest a species-specific variety of fucosylated glycans in cells and EVs.

There was almost no detection of MAL II and SNA in eECs after digestion with neuraminidase, suggesting the presence of α2,3-/2,6-sialoglycans. De-sialylation of eECs resulted in the internalization of eAMC-EV only in Con A-stained cells, suggesting a key role of sialo- and fucose–glycan receptors in the EV uptake.

When sialic acids are cleaved from eAMC-EVs, there is no uptake by eECs. This demonstrates that sialoglycans crucially contribute to the uptake of eAMC-EVs by eECs. Sialic acid presence represents a hallmark of the conserved glycan profile identified for EVs and thus may constitute an essential element in EV endocytosis [20]. The cleavage of terminal sialic acid residues has been shown to affect the distribution of the EVs [16]. This effect could be related to the modification of the negative charge and the steric hindrance of the glycocalyx [33]. However, the effects of de-sialylation on EV uptake depend on the cell type. Treatment with neuraminidase of MLP29 EVs caused their increased uptake by several cell lines, whereas depressed uptake between MLP29 (mouse liver progenitor) EVs and the NCI lung lines [11]. Also, in ovarian cancer cells, the removal of NeuAc led to a non-significant increase in uptake [30].

It is important to highlight a cytochemical aspect detected during the study: a weak AAL affinity of the nuclei of untreated (normal) eECs was observed (not shown). De-fucosylation did not allow staining of the nuclei of numerous eECs with DAPI. This result suggests some involvement of fucosylated glycans in nuclear activity. It has been reported that bovine endothelial cells and malignant cultured human A431 cells express nuclear and extranuclear nucleolin containing fucose O-glycoepitope [45]. Recently, fucosylated proteins, which have key roles in regulating gene expression in the nucleus of plants, have been described [46].

The negative effects of neuraminidase treatment on DAPI staining of the nuclei of eECs are difficult to explain. However, the presence of sialic acid in nuclei has been demonstrated [47]. The presence of a high level of sialic acid in cell line cultures promotes increased cell density and viability [48]. Lastly, it has been reported that sialic acid removal from the surface of human induced pluripotent stem cells (hiPSCs) by neuraminidase treatment causes an altered cell morphology, cell detachment, and cell differentiation [49].

## 4. Materials and Methods

### 4.1. Materials

All reagents in this study were acquired from Sigma-Aldrich (Milan, Italy) unless otherwise stated. Plastic materials were acquired from Euroclone (Milan, Italy).

Three placentas were collected from three broodmares at the term of a normal pregnancy. All procedures were performed according to standard veterinary practice and conforming to the 2010/63 EU directive on animal protection.

### 4.2. Experimental Design

A preliminary study included the isolation of equine epithelial endometrial cells (eEC) and EVs from eAMCs. Afterward, 5 experiments were carried out: (1) evaluation of the incorporation eAMC-EVs into eECs; (2) characterization of AMC-EV glycans with a panel of FITC-labeled lectins; (3) detection of N-linked, fucosylated, and sialylated glycans on eEC surface with FITC-labelled lectins; (4) effects of de-fucosylation and de-sialylation on the uptake of eAMC-EV by eECs; and (5) evaluation of eAMC-EV incorporation into eECs after masking of carbohydrate residues with unconjugated lectins.

#### 4.2.1. Preliminary Phase

##### Epithelial Endometrial Cell Isolation (eECs) and Culture

Three fresh uteri were recovered from a horse slaughterhouse and then sent to the laboratory on ice. The mares were intended for human consumption and had been killed for reasons unrelated to our studies. Uteri selected for endometrial culture were collected from post-pubertal mares in the early/mild diestrus phase of estrus with an evident corpus luteum on the ovary and no signs of genital disease.

In the laboratory, eEC isolation and primary culture were performed as explained in the protocol described by Perrini et al. [4]. Briefly, uterine horns ipsilateral to the corpus luteum were opened, and the endometrium was washed with sterile saline, which was then mechanically detached from the underlying tissues with scissors. Isolated endometrium was cut into strips and digested in sterile-filtered Hank’s buffered salt solution (supplemented with 1 mg/mL collagenase II, 4 mg/mL bovine serum albumin, and 0.4 mg/mL DNase I) for 3 h at 38.5 °C in a shaking bath. Endometrial cells were subsequently passed through an 80 μm pore size filter, centrifuged at 200× *g* for 10 min, and then subjected to two washes in phosphate buffer solution (PBS).

A primary culture was initiated using a complete medium composed of HG-DMEM supplemented with 10% FBS, 0.25 μg/mL amphotericin B, penicillin (100 UI/mL), streptomycin (100 μg/mL), and 2 mM L-glutamine. Before seeding, the total number of live cells was counted using the Trypan blue exclusion method and a Bürker chamber.

Cells obtained after digestion were plated at a density of 1 × 10⁵ cells/cm^2^ and incubated in a humidified atmosphere, 5% CO_2_ at 38.5 °C. After 18 h of incubation, endometrial connective cells were adherent to the bottom of the flask, while endometrial epithelial cells remained in suspension. So, at this time, the culture medium rich in the epithelial cell population was removed and seeded again. In this way, connective and epithelial cells were obtained separately. Once sub-confluency was reached, both cell populations were detached using 0.05% trypsin-EDTA and then cryopreserved at passage 1.

Only epithelial endometrial cells were used in this study.

##### Equine Amniotic Membrane Collection and Amniotic Mesenchymal Stromal Cell (eAMC) Isolation

Three allanto-amniotic membranes were collected after a physiological delivery. All procedures were performed according to standard veterinary practice and conforming to the 2010/63 EU directive on animal protection. Allanto-amniotic membranes were transported at 4 °C and processed within 8 h from collection.

Amniotic membranes were mechanically detached from the allantois and cut into fragments that were kept in PBS containing 2.4 U/mL dispase (Becton Dickinson & Company, Milan, Italy) for 9 min at 38.5 °C. Endometrial sections were then incubated in HG-DMEM (supplemented with 10% fetal bovine serum (FBS) and 2 mM L-glutamine, Euroclone) for 5–10 min at room temperature and digested with 1 mg/mL collagenase type I and 20 µg/mL DNase (Roche, Mannheim, Germany) for 3 h at 38.5 °C. The products of enzymatic digestion were filtered with a 100 µm filter. Equine AMCs were collected by centrifugation at 250× *g* for 10 min.

##### Equine Amniotic Mesenchymal Stromal Cell (eAMC) Expansion and Conditioned Medium Production

Isolated eAMCs were seeded in a complete medium enriched with 10 ng/mL epidermal growth factor at a density of 1 × 10^5^ cell/cm^2^ for the first passage and at 1 × 10^4^ cell/cm^2^ for subsequent passages. Cells were cultured in a controlled atmosphere with 5% CO_2_ and 90% humidity at 38.5 °C until passage 3 (P3). Cellular viability was assessed using Trypan blue staining.

To produce conditioned medium, eAMCs at P3 were cultured in a serum-free ultraculture medium (Ultraculture, Lonza, Milan, Italy) for 3 days. Each morning, conditioned media were collected and replaced with fresh media, then centrifuged at 1600× *g* for 20 min to remove cells and at 4500× *g* for 20 min to remove debris. Finally, CM was stored at −80 °C until EV isolation.

##### Isolation of Extracellular Vesicles (EVs) and Their Labelling

To isolate EVs, the conditioned medium was subjected to ultracentrifugation at 100,000× *g* (Beckman Coulter OptimaX, Milan, Italy), 4 °C for 1 h. After pellet resuspension in serum-free medium and assessment of EV size and concentration through Nanosight Instrument, EV aliquots were prepared and stored at −20 °C until use. When necessary, EVs were thawed at 38.5 °C in a water bath.

The characterization of EVs was performed according to MISEV guidelines [24], as reported by Lange-Consiglio et al. [23], as the isolation of eAMCs, the production of CM, and the isolation of EVs is largely standardized in this laboratory. In addition, microRNA cargo of eAMCs and their EVs had also been previously analyzed [25].

In this study, EVs were characterized only by a Nanosight Instrument to detect their concentration and size.

To detect the EV uptake by eECs through fluorescent microscopy, EVs with PKH-26 were labeled. This fluorescent compound is a red aliphatic chromophore that, intercalating into the lipid bilayer of the vesicles, marks them in red. After EV ultracentrifugation, the resulting pellet was resuspended in 1 mL of the reagent supplied by the kit. A volume of 4 µL of fluorochrome was added, and the suspension was incubated at 38.5 °C, 5% CO_2_ for 30 min in the dark. A volume of 7 mL serum-free DMEM was added to stop the reaction. The suspension was subjected to another ultracentrifugation at 100,000× *g*, 1 h at 4 °C, and the final pellet was resuspended in serum-free medium and stored at −20 °C.

#### 4.2.2. Evaluation of eAMC-EV Incorporation into eECs

To control the incorporation of eAMC-EVs into eECs, cells were seeded on culture slides (13 mm; Nalgen Nunc International, Rochester, NY, USA) and collocated into 24 well plates at a density of 60 × 10^3^. At 60% of confluence, cells were co-cultured for 24 h with 400 × 10^6^ eAMC-EVs/mL labeled with PKH-26 dye. Fifteen minutes before this time point, eECs were incubated with 10 μg/mL Hoechst 33343 for nuclear staining. Equine AMC-EV uptake was evaluated using an Olympus BX51 microscope provided with a Scion Corporation 1394 video camera combined with software for image acquisition and analysis (Image-Pro Plus 5.1-Media Cybernetics, Immagini & Computer, Bareggio, Italy). The excitation wavelength for PKH-26 was set at 550 nm, while 567 nm was the emission wavelength. Hoechst 33342 dye was excited at 353–365 nm while the emission wavelength was set at 460 nm.

A Leica SP2 laser scanning confocal microscope (Leica Microsystems Srl, Buccinasco, Italy) equipped with a PL Fluotar 20× AN 0.5 Dry objective was used to assess the internalization of eAMC-EVs.

#### 4.2.3. Characterization of AMC-EV Glycans with a Panel of FITC-Labeled Lectins

To analyze the glycan pattern of the eAMC-EVs, a panel of nine FITC-labeled lectins (Table 1) and PKH-26-labeled and unlabeled eAMC-EVs were used. MAL II lectin was purchased from Glycomatrix (Dublin, OH, USA), while all the other lectins were acquired from Vector Laboratories (Burlingame, CA, USA).

PKH-26-labeled and/or unlabeled eAMC-EVs were smeared on poly-L-lysine coated glass slides and air-dried overnight at room temperature (RT) in the dark. They were then incubated in a humid chamber at RT for 1 h in the dark with the lectins diluted in 0.01 M PBS (pH 7.4) at the concentration given in Table 1. Subsequently, the slides were washed in PBS and mounted in Vectashield Antifade Mounting Medium with DAPI (Vector Lab., Burlingame, CA, USA). Slides were observed and photographed at 60× magnification under a light photomicroscope (Eclipse Ni-U, Nikon, Tokyo, Japan) equipped with a digital camera (DS-U3, Nikon, Japan) and analyzed using the image-analyzing program MIS Elements BR (Ver. 4.20) (Nikon, Japan). The FITC fluorochrome is activated at a wavelength of 490 nm and emits at 520 nm. Each experiment was repeated three times for each sample.

#### 4.2.4. Detection of N-Linked, Fucosylated, and Sialylated Glycans on eEC Surface with FITC-Labelled Lectins

Equine ECs were seeded on culture slides collocated in 24 well plates at a density of 60 × 10^3^. At 60% of confluence, the slides were rinsed with 0.05 M PBS. To detect the N-linked, fucosylated, and sialylated glycans, eECs were fixed in 4% (v/v) buffered paraformaldehyde, pH 7.4, for 1 h at RT and, after three rinsings in PBS, they were incubated at RT for 1 h in the dark with the lectins Con A, AAL, MAL II, and SNA diluted in PBS (pH 7.4) at the same concentration used for eAMC-EVs glycoanalysis (Table 1). Slides were then rinsed in the same buffer and mounted in Vectashield Antifade Mounting Medium with DAPI (Vector Laboratories, Burlingame, CA, USA).

Lectin staining controls were represented by the replacement of substrate medium with buffer in the absence of lectins and by the incubation with each lectin in the presence of its hapten sugar.

Slides were observed under a light photomicroscope (Eclipse NieU, Nikon, Japan) equipped with a digital camera (DS-U3, Nikon, Japan) and photographed at 60× magnification.

The same procedure was performed for eECs co-cultured for 24 h with 400 × 10^6^ eAMC-EVs/mL (dose–response curve not shown) labeled with PKH-26 dye.

Each experiment was repeated three times for each sample.

#### 4.2.5. Effect of De-Fucosylation and De-Sialylation on eEC Uptake by eAMC-EVs

Equine ECs seeded on culture slides, and eAMC-EVs were both treated with fucosidase or neuraminidase. Hydrolysis of fucose from eECs and eAMC-EVs was carried out for 3 h at 38.5 °C by the addition of α1,2-fucosidase (0.004 units/mL in Tris/HCl 20 mM, pH 7.5) (Sigma-Aldrich).

Removal of sialic acids from eECs and eAMC-EVs was performed by digestion at 38.5 °C for 1 h with 0.86 U/mL of neuraminidase type X from *Clostridium perfringens* (Sigma-Aldrich) in 0.1 M sodium acetate buffer (pH 5.5) containing 10 mM CaCl2. Subsequently, eECs and eAMC-EVs were stained with the FITC-conjugated MAL II and SNA.

Enzyme-treated eECs and eAMC-EVs were co-incubated for 24 h at 38.5 °C. The timeline of the experiment is schematized in Figure 9.

*Five hours before* the 24 h of co-incubation, 90 million eAMC-EVs, previously labeled with PKH-26, were incubated with 200 µL of fucosidase solution. *Three and half hours* before the start of 24 h of co-incubation, 200 µL of neuraminidase solution was added to eAMC-EVs for 1 h. After both treatments, eAMC-EV samples were diluted with PBS, ultracentrifuged at 100,000× *g*, 1 h, at 4 °C, and serum-free medium was used to resuspend the pellet.

*Three hours before* the co-incubation, a sample of eECs at 60% of confluence was washed with PBS, treated with 200 µL of fucosidase, and incubated at 38.5 °C for 3 h in a 5% CO_2_ incubator. *One hour* before the start of the 24 h co-incubation, another sample of eECs was treated with 200 µL of neuraminidase for 1 h.

After the de-fucosylation and de-sialylation treatments of both cells and EVs, eECs and eAMC-EVs were incubated in all combinations for 24 h. Slides were then fixed with 10% formalin for 1 h.

Controls are represented by eECs not treated with fucosidase or neuraminidase and incubated with eAMC-EV labeled with PKH-26.

Each experiment was repeated three times for each sample.

#### 4.2.6. Evaluation of eAMC-EV Incorporation into eECs After Masking of Carbohydrate Residues with Unconjugated Lectins

The experiment involved the incubation of eECs with PKH-26-labeled eAMC-EVs previously reacted with the 9 unconjugated lectins mentioned above. This study was carried out in three steps.

Step 1: PKH-26 labeled EVs were incubated in suspension with each of the unconjugated lectins overnight at room temperature. The following day, eAMC-EVs were ultracentrifuged to remove the excess of unbound lectins, and the total volume was brought to 1 mL with a serum-free DMEM base medium.Step 2: forty-eight hours before the co-incubation experiment, eECs at P1 were thawed and seeded on 24 × 24 coverslips placed in 9 wells of six-well plates (one for each lectin used). Cells were cultured with complete DMEM.Step 3: labeled eAMC-EVs bound to the 9 unconjugated lectins were added to eECs (one lectin per well).

Fifteen minutes before the end of 24 h of co-incubation at 38.5 °C with 5% CO_2_, Hoechst was added to the culture medium to allow observation of eEC nuclei under fluorescence microscopy. Once the co-incubation was concluded, wells were washed with PBS and fixed with 10% formalin. Finally, a further wash with PBS was performed, and the AMC-EV uptake by eECs was observed using fluorescence microscopy.

### 4.3. Statistical Analysis

The fluorescence signal intensity for 15 cells binding each FITC-labelled lectin from untreated (normal) and de-glycosylated eECs and eAMC-EVs was measured with background subtraction. The images were analyzed using the NIS Elements BR (Vers. 4.20) (Nikon, Japan) image-analyzing program. Values were expressed as means ± standard deviation (S.D.). The results were evaluated for statistical significance using the Student’s *t*-test.

## 5. Conclusions

This study showed for the first time that the internalization of EVs by eECs occurs through the interaction between the surface glycans of both EVs and eECs. Furthermore, lectin reactivity and de-glycosylation treatments demonstrated that fucose- and sialic acid-containing glycoproteins on the surface of eECs and eAMC-EVs play a crucial role in EV uptake. The evidenced α1,2-linked fucose on the surface of both eECs and eAMC-EVs could represent an equine biomarker. Therefore, studies should be conducted on other equine cell lines and EVs to confirm or refute the presence of α1,2-fucosylated glycans and to define their role in the interaction between EVs and target cells.

## Figures and Tables

**Figure 1 ijms-26-01784-f001:**
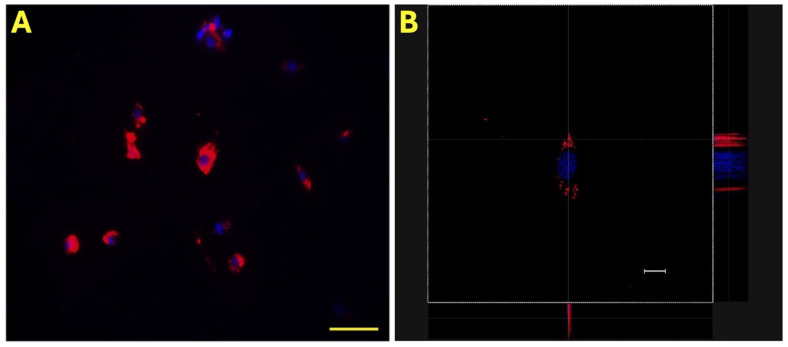
Internalization of eAMC-EVs. Representative micrographs of eEC internalization of PKH-26-labeled eAMC-EVs under fluorescence microscopy (**A**) and under confocal microscopy (**B**). eAMC-EVs are red, and endometrial cell nuclei are blue. Scale bar: (**A**), 20 µm; (**B**): 50 µm.

**Figure 2 ijms-26-01784-f002:**
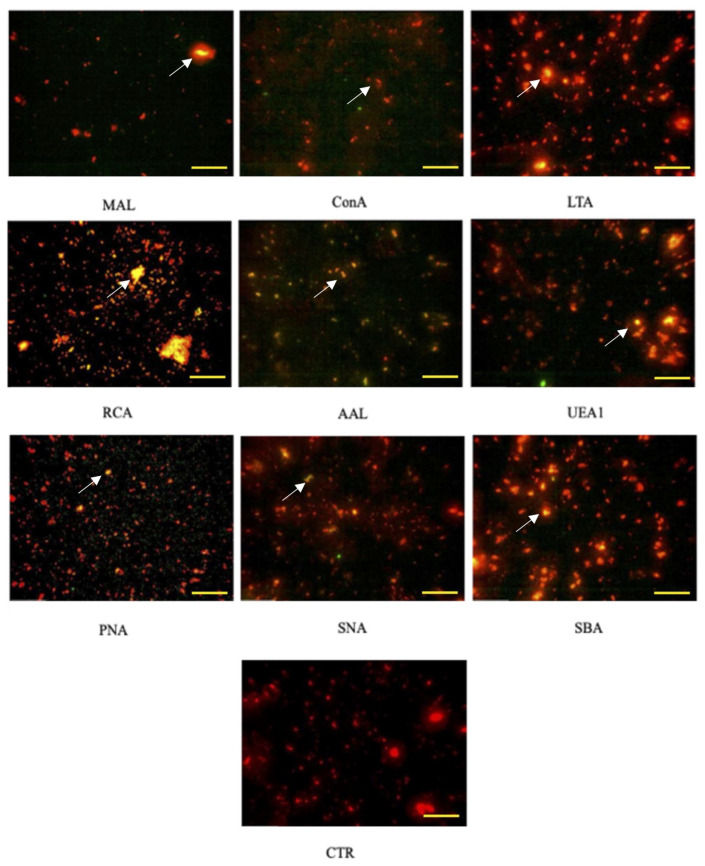
Positivity of AMC-EVs to lectins. Each microphotograph shows the positivity to one of the 9 lectins studied (indicated by the arrows). Note the absence of the yellow/orange color in EVs not incubated with lectins (CTR). CTR, control. The bar corresponds to 20 µm.

**Figure 3 ijms-26-01784-f003:**
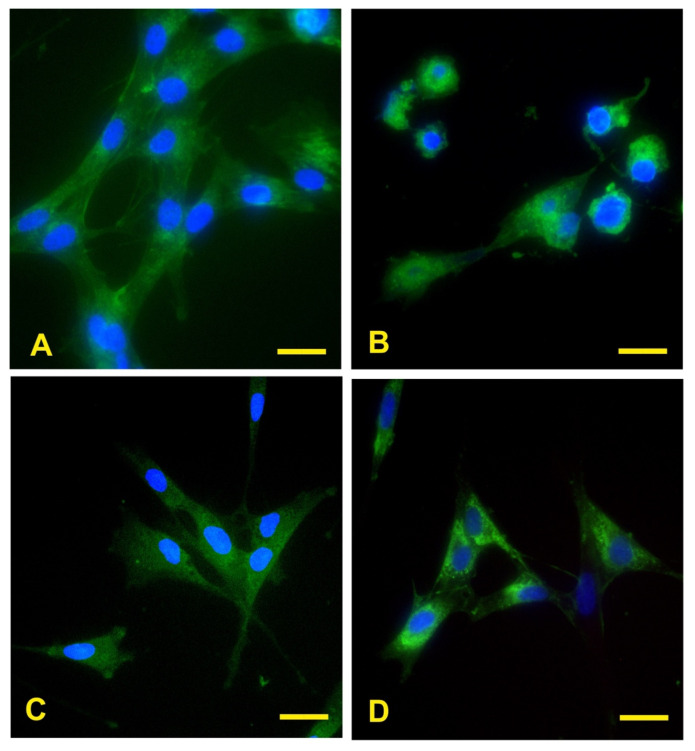
Reactivity of ECs to lectins. The green fluorescence demonstrates eECs reactivity for AAL, Con A, MAL II, and SNA reactivity for eECs (**A**–**D**). Scale bar: 25 μm.

**Figure 4 ijms-26-01784-f004:**
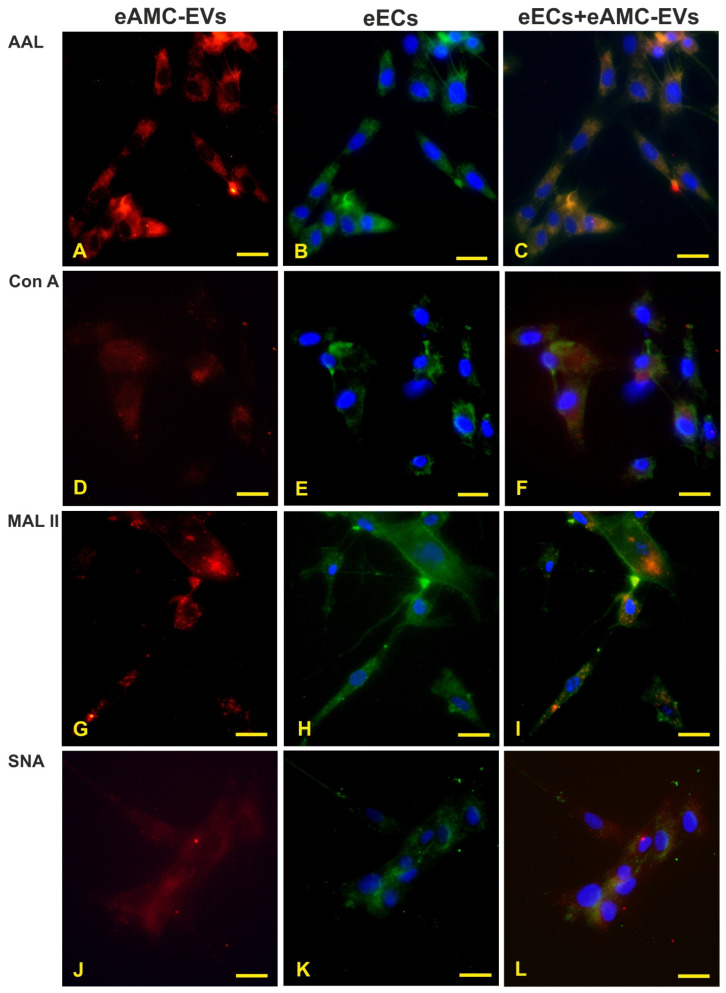
Internalization of eAMC-EVs before and after reactivity to lectins. Presence of eAMC-EVs within eECs before and after staining with AAL, Con A, MAL II, and SNA. (**A**,**D**,**G**,**J**) eAMC-EVs (red color) in the cytoplasm of eECs whose lectin reactivity is displayed in (**B**,**E**,**H**,**K**) (green color). In the merged pictures (**C**,**F**,**I**,**L**), eAMC-EVs internalized by eECs appear in orange. Scale bar: 25 μm.

**Figure 5 ijms-26-01784-f005:**
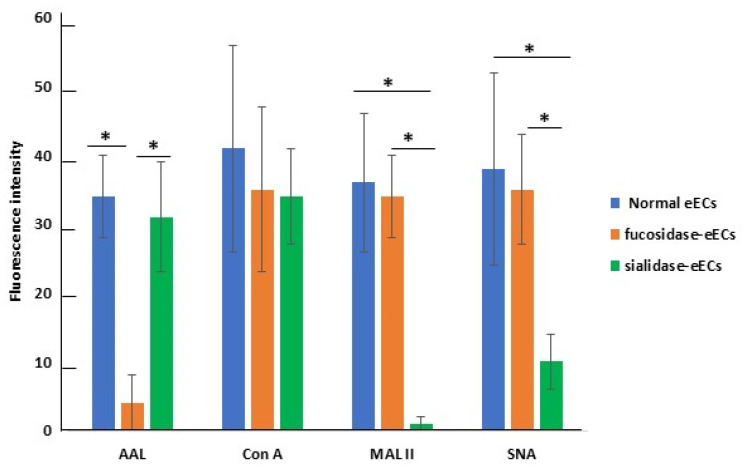
Quantification of FITC signals in eECs. The values are expressed as means ± standard deviation (S.D.). Asterisks indicate statistically significant differences (*p* < 0.05).

**Figure 6 ijms-26-01784-f006:**
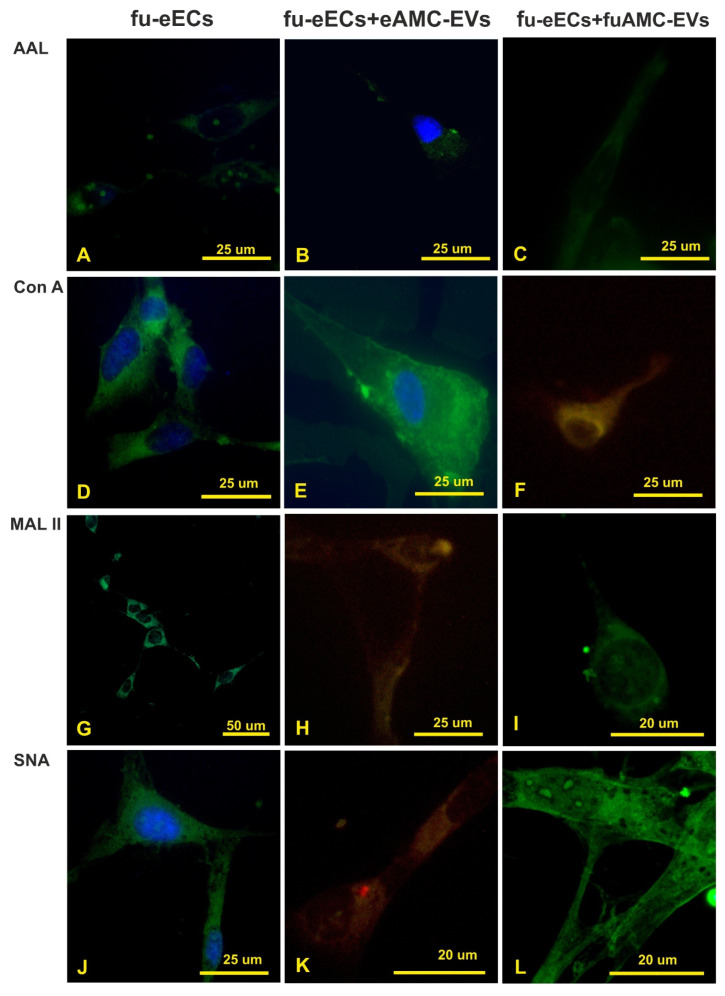
Treatment with fucosidase. Effect of fucosidase treatment on AAL, Con A, MAL II, and SNA reactivity of eECs (**A**,**D**,**G**,**J**); uptake of eAMC-EVs after fucosidase treatment of eECs (**B**,**E**,**H**,**K**); effect of fucosidase treatment on eECs and eAMC-EVs on their incorporation by eECs (**C**,**F**,**I**,**L**). Note that fucose removal strongly affects the uptake of eAMC-EVs by eECs (compare with Figure 5).

**Figure 7 ijms-26-01784-f007:**
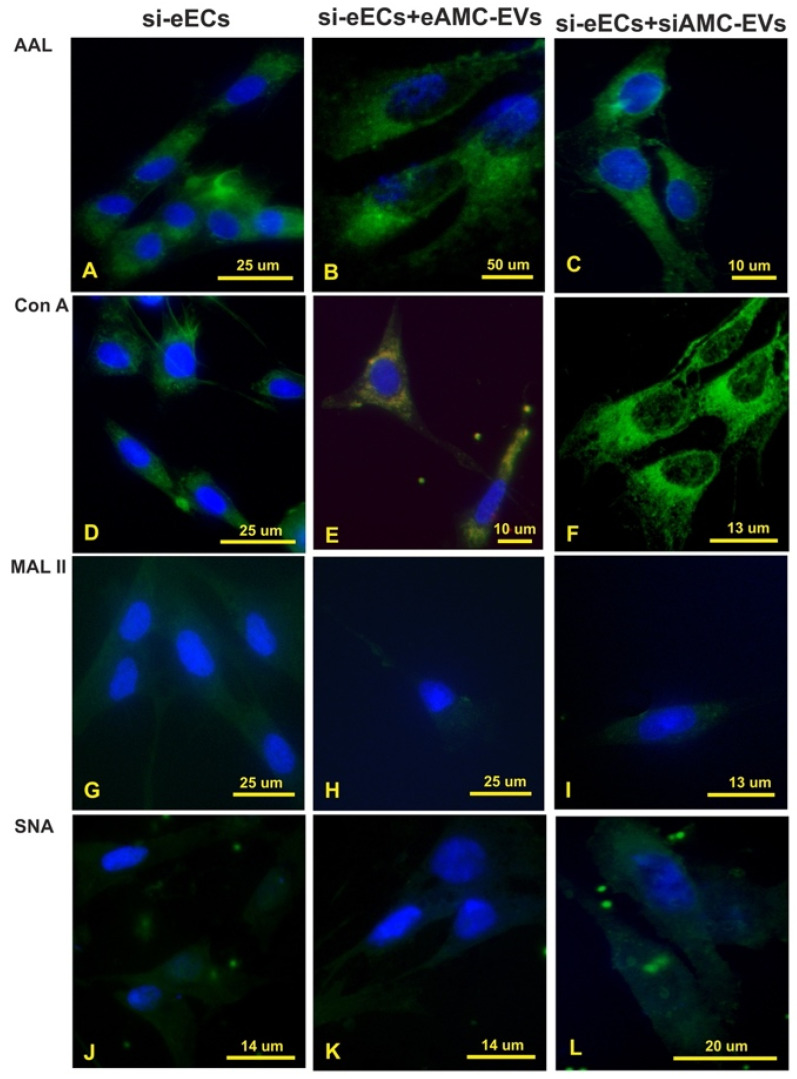
Treatment with neuraminidase. Effect of neuraminidase treatment on AAL, Con A, MAL II, and SNA reactivity of eECs (**A**,**D**,**G**,**J**); uptake of eAMC-EVs after neuraminidase treatment of eECs (**B**,**E**,**H**,**K**); effect of neuraminidase treatment on eECs and eAMC-EVs on their incorporation by eECs (**C**,**F**,**I**,**L**). Note that sialic acid removal strongly affects the uptake of eAMC-EVs by eECs (compared with Figure 5).

**Figure 8 ijms-26-01784-f008:**
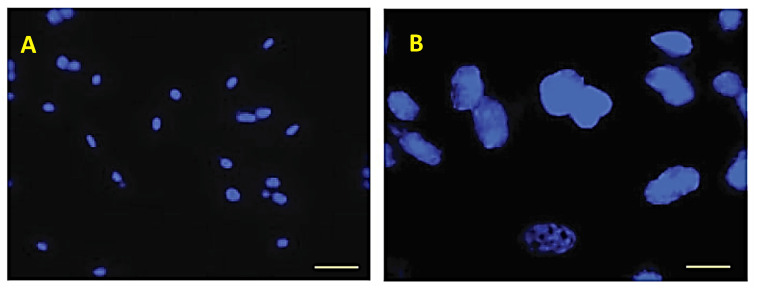
Incorporation into eECs after masking of carbohydrate residues with unconjugated lectins. Micrographs of Hoechst-labeled nuclei at two different magnifications. No incorporation of labeled eAMC-EVs is visible. Scale bar: (**A**) 20 µm; (**B**) 10 µm.

**Figure 9 ijms-26-01784-f009:**
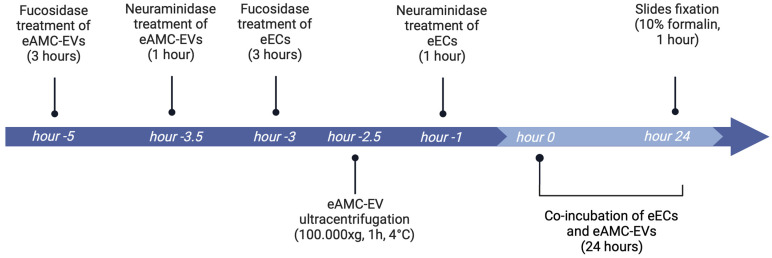
Reference timeline of the experiment (created with Biorender.com).

**Table 1 ijms-26-01784-t001:** Lectins used and their sugar specificity. The symbol nomenclature comes from Carbohydrate Structure Databases (http://csdb.glycoscience.ru/database/core/wizard.html (accessed on 16 March 2024)).

Lectin Abbreviation	Source of Lectin	Concentration (µg/mL)	Typical Glycan Bound	Inhibitory Sugar
AAL	*Aleuria aurantia*	50	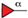 α1-2/1-3/1-4/1-6	Fuc
Con A	*Canavalia ensiformis*	50	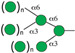	Man
LTA	*Lotus tetragonolobus*	50	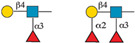	Fuc
MAL II	*Maackia amurensis*	50	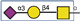	NeuNAc
PNA	*Arachis hypogaea*	50	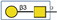	Gal
RCA_120_	*Ricinus communis*	50	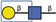	Gal
SBA	*Glycine max*	50	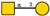	GalNAc
SNA	*Sambucus snigra*	50	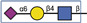	NeuNAc
UEA I	*Ulex europaeus I*	50	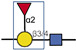	Fuc
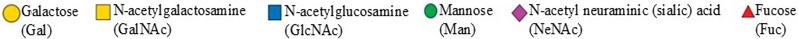

## Data Availability

The dataset supporting the conclusions of this article is included within the article (and there are no additional files).

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
