# Peer review of "Role of Glycans in Equine Endometrial Cell Uptake of Extracellular Vesicles Derived from Amniotic Mesenchymal Stromal Cells"

_ijms, 2025, doi:10.3390/ijms26041784_

Round 1
Reviewer 1 Report
Comments and Suggestions for Authors
Comments and Suggestions:
Title: Role of glycans in equine endometrial cell uptake of extracellular vesicles derived from amniotic mesenchymal stromal cells.
Reviewer’s report:
The manuscript by Gaspari et al., is an interesting study which describes about the internalization of EVs derived from amniotic mesenchymal/stromal cells (eAMC-EVs) by equine endometrial cells (eECs) which is influenced by glycan profiles. They found that N-linked high manose glycans, fucosylated and sialylated glycans expressed by both eECs and eAMC-EVs were involved in the uptake of EVs and uptake was decreased by the cleavage of α1,2-linked fucose and α2,3/α2,6-linked sialic acids. They concluded that fucosylated and sialylated glycans are highly relevant in the transfer of bioactive molecules with effects on regenerative medicine treatments.
Major Points:
1. Introduction: please merge small paragraphs and make it total of two paragraphs. Please do the same in whole manuscript
2. Figure 1: why the PKH26 fluorescence is seen very less in case of confocal microscopy as compared to fluorescence microscopy?
Minor Points:
1. Points 2.1, 2.2, 2.3, 2.4, 2.5: The word “Experiment 1” and so on is not needed.
2. Figure 4: on the top of B, E, H, K; please mention eECs with lectins.
3. Figure 5: the significance line in AAL should be drawn between normal-eECs and sialidase-eECs.
Comments on the Quality of English LanguageThe paragraph arrangement and English grammar should be improved significantly.
Author Response
The manuscript by Gaspari et al., is an interesting study which describes about the internalization of EVs derived from amniotic mesenchymal/stromal cells (eAMC-EVs) by equine endometrial cells (eECs) which is influenced by glycan profiles. They found that N-linked high manose glycans, fucosylated and sialylated glycans expressed by both eECs and eAMC-EVs were involved in the uptake of EVs and uptake was decreased by the cleavage of α1,2-linked fucose and α2,3/α2,6-linked sialic acids. They concluded that fucosylated and sialylated glycans are highly relevant in the transfer of bioactive molecules with effects on regenerative medicine treatments.
ANSWER: the authors thank the Reviewers for consideration, kind comments, and helpful suggestions. According to the comments and suggestions, we have carefully evaluated all critical points, and the manuscript has been thoroughly revised.
The authors hope that the manuscript is now suitable for publication.
Major Points:
- Introduction: please merge small paragraphs and make it total of two paragraphs. Please do the same in whole manuscript
ANSWER: the authors thank the reviewer for her/his suggestion and the paragraphs have been condensed.
- Figure 1: why the PKH26 fluorescence is seen very less in case of confocal microscopy as compared to fluorescence microscopy?
ANSWER: the authors thank the reviewer for her/his suggestion and Figure 1 has been replaced.
Minor Points:
- Points 2.1, 2.2, 2.3, 2.4, 2.5: The word “Experiment 1” and so on is not needed.
ANSWER: the authors thank the reviewer for her/his suggestion and this word has been removed through all the manuscript
- Figure 4: on the top of B, E, H, K; please mention eECs with lectins.
ANSWER: the authors thank the reviewer for her/his suggestion and Figure 4 has been modified
- Figure 5: the significance line in AAL should be drawn between normal-eECs and sialidase-eECs.
ANSWER: the authors thank the reviewer for her/his suggestion and Figure 5 has been modified
We feel that we have addressed all queries raised by the referees and hope that the paper is now acceptable for consideration by the Editor of International Journal of Molecular Science.
We thank you in advance for your time and consideration.
On behalf of all authors best regards,
Giulia Gaspari
Reviewer 2 Report
Comments and Suggestions for Authors
Comments and Suggestions for Authors
This study investigates the role of surface glycans in the uptake of extracellular vesicles (EVs) by equine embryonic cells (eECs). The study found that surface glycans are involved in the internalization of amniotic mesenchymal/stromal cells by eECs, and that fucosylated and sialylated glycans are highly relevant in the transfer of bioactive molecules. EVs derived from amniotic mesenchymal/stromal cells are internalized by eECs, which can have positive effects on regenerative medicine treatments.
The article is well-structured, with clearly described methods and satisfactory results supported by figures. The discussion effectively integrates findings from previously published studies, strengthening the research and its presented results. However, there are some fundamental comments that need to be addressed.
Minor comment
1. In Figure 2, point arrows at the microphotographs showing positivity to lectins to emphasize differences.
2. Figure 3 requires a more detailed legend for A, B, C, and D.
3. Figure 5 needs some refinement. Ensure the X and Y axes are visible.
4. Provide a suitable reference for lines 181-182.
5. Minor English editing is required to maintain the flow, particularly in the discussion section.
Typo error
1. In line 46, it could be 'also' in 'glycans lso play'.
2. Place a full stop at the end of line 297.
3. Ensure a consistent format for 'hours' throughout the manuscript.
4. Check the reference pattern and correct it accordingly.
Author Response
This study investigates the role of surface glycans in the uptake of extracellular vesicles (EVs) by equine endometrial cells (eECs). The study found that surface glycans are involved in the internalization of amniotic mesenchymal/stromal cells by eECs, and that fucosylated and sialylated glycans are highly relevant in the transfer of bioactive molecules. EVs derived from amniotic mesenchymal/stromal cells are internalized by eECs, which can have positive effects on regenerative medicine treatments.
The article is well-structured, with clearly described methods and satisfactory results supported by figures. The discussion effectively integrates findings from previously published studies, strengthening the research and its presented results. However, there are some fundamental comments that need to be addressed.
ANSWER: the authors thank the Reviewers for consideration, kind comments, and helpful suggestions. According to the comments and suggestions, we have carefully evaluated all critical points, and the manuscript has been thoroughly revised.
The authors hope that the manuscript is now suitable for publication.
Minor comment
In Figure 2, point arrows at the microphotographs showing positivity to lectins to emphasize differences.
ANSWER: the authors thank the reviewer for her/his suggestion and arrows have been added in the Figure
- Figure 3 requires a more detailed legend for A, B, C, and D.
ANSWER: the authors thank the reviewer for her/his suggestion and the legend of Figure 3 has been modified
- Figure 5 needs some refinement. Ensure the X and Y axes are visible.
ANSWER: the authors thank the reviewer for her/his suggestion and the axes have been added
- Provide a suitable reference for lines 181-182.
ANSWER: the authors thank the reviewer for her/his suggestion and the references has been added (citation 10: Williams et al., 2018)
- Minor English editing is required to maintain the flow, particularly in the discussion section.
ANSWER: the authors thank the referee for her/his comment, but the paper was proofread for English by an English mother tongue prior to submission but carefully revised now. This native speaker is a veterinary who was also co-editor of Small Animal Practice and vice-president of British Small Animal Veterinary Association.
Her name is
Katie McConnell (MA VetMB CertVR CertSAM MRCVS)
km.copyed@gmail.com
Typo error
- In line 46, it could be 'also' in 'glycans lso play'.
- Place a full stop at the end of line 297.
- Ensure a consistent format for 'hours' throughout the manuscript.
ANSWER: the authors thank the referee for her/his suggestions and the format of “hours” has been uniformed
- Check the reference pattern and correct it accordingly.
ANSWER: the authors thank the referee for her/his suggestion and the references have been checked
We feel that we have addressed all queries raised by the referees and hope that the paper is now acceptable for consideration by the Editor of International Journal of Molecular Science.
We thank you in advance for your time and consideration.
On behalf of all authors best regards,
Giulia Gaspari